# Active Bio-Based Pressure-Sensitive Adhesive Based Natural Rubber for Food Antimicrobial Applications: Effect of Processing Parameters on Its Adhesion Properties

**DOI:** 10.3390/polym13020199

**Published:** 2021-01-07

**Authors:** Theerarat Sengsuk, Ponusa Songtipya, Ekwipoo Kalkornsurapranee, Jobish Johns, Ladawan Songtipya

**Affiliations:** 1Center of Excellence in Bio-Based Materials and Packaging Innovation, Faculty of Agro-Industry, Prince of Songkla University, Hat-Yai 90110, Thailand; Moo-kwan_psu@hotmail.com (T.S.); ponusa.j@psu.ac.th (P.S.); 2Division of Physical Science, Faculty of Science, Prince of Songkla University, Hat-Yai 90110, Thailand; ekwipoo.k@psu.ac.th; 3Research and Development Center, Department of Physics, Raja Rajeswari College of Engineering, Bangalore 560074, India; jobish_johns@rediffmail.com

**Keywords:** bio-based pressure-sensitive adhesive, natural rubber, xyloglucan, cinnamon oil, antimicrobial packaging, processing parameters

## Abstract

A novel active bio-based pressure-sensitive adhesive incorporating cinnamon oil (Bio-PSA/CO) obtained from the mixture of natural rubber (NR), xyloglucan (XG), and cinnamon oil (CO) for food antimicrobial applications were successfully developed by using a two-roll mill mixer. The effect of the main process factors (i.e., nip gap and mastication time) and XG content on the adhesion properties of the obtained PSA were investigated with different coated substrates including kraft paper, nylon film, polypropylene (PP) film, and aluminum foil (Al). The results suggested that the developed NR-PSA/CO could be applied well to all types of substrate materials. Peel strength and shear strength of the NR-PSA/CO with all substrate types were in the ranges of ~0.03 × 10^2^–5.64 × 10^2^ N/m and ~0.24 × 10^4^–9.50 × 10^4^ N/m^2^, respectively. The proper processed condition of the NR-PSA/CO was represented with a nip gap of 2 mm and a mastication time of 15 min. An increase in XG content up to 40–60 phr can improve the adhesion properties of the adhesive. The resulting material could be used as an active sticky patch to extend the shelf-life of food in a closed packaging system. The shelf-life of the food samples (banana cupcake) could be extended from 4 to 9 days with NR-PSA/CO patch.

## 1. Introduction

Recently, many antimicrobial active packaging systems have been developed in order to extend the shelf life of food, improve food safety, and maintain the quality of packed food. Several antimicrobial compounds such as silver nanoparticles, zeolite, ethanol, and chlorine dioxide [1] have been added into packaging or packaging materials in order to inhibit the growth of microorganisms which is the main cause of food spoilage. However, consumers still have questions about the safety of these chemicals that might migrate into food. Therefore, the use of essential oils that can be extracted from natural products as antimicrobial substances for fresh chicken drumsticks [2], fruits [3], vegetables [4], bread [5], and cake [6] have been studied more, due to their non-toxic and high antimicrobial activity by disruption of the bacterial cell membrane [7,8]. For prolonging the shelf-life of food by using essential oils, several types of essential oils have been entrapped into carrier materials like starch [3], silica gel [9], ß-cyclodextrin [10], and polyethylene (HDPE) resins [5] before packing the active particles into a sachet (a small porous bag). A close system of packed food with this sachet showed the reduction of spoilage bacteria growth because the vapor of the essential oils can be released and then expose to the food surface. However, this procedure often causes a change of flavor if the active sachet comes into direct contact with the food. Inserting the sachet into the food packaging may cause consumers to misunderstand that it can be eaten. From this point, the new idea for preparing an antimicrobial sticky patch called “active bio-based pressure-sensitive adhesives patch incorporating cinnamon oil (CO) (Bio-PSA/CO)” has been proposed in this work as shown in Figure 1. This is due to the pressure-sensitive adhesives (PSA) as an interesting material that can be permanently tacked with substrates and it does not require any external activation for tacking (i.e., moisture, solvent, and heat). Moreover, it is sufficient cohesiveness, elasticity, and ease of removal without any residual adhesives on the substrates [11,12] making the system more attractive in food packaging applications.

Normally, the majority of PSAs were prepared from the blends of synthetic rubber (e.g., styrene-isoprene block copolymer (S-I-S) [13], polyisobutylene (PIB) [14], butyl rubber (BR) [15], and styrene butadiene rubber (SBR) [12] by using hot melt mixing process. However, their production cost is found to be still high and the obtained PSAs are non-biodegradable. Therefore, in order to replace synthetic rubber, the PSA based-natural rubber (NR) has been developed because of its non-toxic nature and good tacking with many types of substrate materials especially when its molecular weight is being reduced [16,17,18,19,20]. Moreover, it can be used as a bio-device for a controlled release system [21]. In solid natural rubber (NR) systems, several types of tackifiers such as acrylic resin [16], guar gum, and aqua gel [17] have been used to promote specific chemical adhesion and/or create lower the softening point of the polymer system. However, a novel tackifier (hydroxyethyl cellulose, HEC) for the PSA-based NR has been recently reported by Kalkornsurapranee, et al. [20]. It is surprising that the increase of the tackiness of the PSA is a result of preventing the relaxation and recombination of NR molecules after mastication as well as increase the surface interaction by adding the HEC. Moreover, the adhesion properties of the PSA can be increased by optimizing the processing parameters including initial viscosity of NR, mastication time, and step of mixing [20].

In this work, the Bio-PSA/CO was developed from NR through a two-roll mill mixer. The XG, hemicellulose extracted from agricultural waste (tamarind seed) was incorporated into the Bio-PSA/CO to act as a tackifier because its low cost, and safe for use in food contact material application. Moreover, CO was used as an active compound due to its superior antimicrobial abilities specifically suitable to inhibit the growth of gram-positive bacteria (e.g., *Staphylococcus aureus*, *Bacillus subtilis*, *Bacillus cereus* and *Listeria monocytogenes*) [22,23,24,25,26], gram-negative bacteria (e.g., *Pseudomonas aeruginosa*, *Escherichia coli*, and *Salmonella typhimurium* [22,23,24], and molds (e.g., *Penicillum expansum* and *Rhizopus nigricans*, *Aspergillus flavus*) [24,27]. In the beginning, the preliminary test of prolonging the shelf-life of food was investigated. The effect of processing procedure factors including nip gap, mastication time, and tackifier content on adhesion properties (i.e., peel strength and shear strength) as well as Mn, Mw, and PDI of the obtained Bio-PSA were examined and discussed in-depth. The adhesion properties of the Bio-PSA patches with several types of food packaging materials including kraft paper, nylon film, polypropylene (PP) film, and aluminum foil (Al) were compared. Moreover, the antimicrobial test of the Bio-PSA patch for a banana cupcake was also investigated.

## 2. Materials and Methods

### 2.1. Materials

Standard Thai Rubber 5L (STR 5L) was manufactured by Chalong Latex Industry Co., Ltd. (Songkhla, Thailand). Tamarind seed powder (TSP) was supported by G.M. IchiHara Co., Ltd. (Pathum Thani, Thailand). Cinnamon oil (CO) (99.99%) was purchased from Lapis Tropical Spa Products Co., Ltd. (Bangkok, Thailand). 95% ethanol with commercial-grade was provided by L.B. Science Limited Partnership (Songkhla, Thailand). Nutrient agar (NA), potato dextrose agar (PDA), nutrient broth (NB), and potato dextrose broth (PDB) were purchased from HiMedia Laboratories Pvt. Ltd. (Mumbai, India). Distilled water was used as an extraction solvent and media preparation.

### 2.2. Preparation of Xyloglucan

The tamarind seed powder (TPS) slurry was first prepared by mechanical mixing of 100 g of TSP with 1000 mL of distilled water. The slurry was heated for 30 min with a temperature of 80–100 °C with a mechanical stirrer of 250 rpm. Further, centrifugation with 7000 rpm at ambient temperature was continued for 20 min to remove proteins and impurities. The obtained supernatant (XG solution) was added into ethanol with a weight ratio of 1:2 in order to precipitate the XG [28]. The solid XG was filtered with a food strainer and dried in a hot air oven under 40 °C. Then, the obtained XG was ground with mortar before sieving through a metal sieve (35 mesh, 500 μm).

### 2.3. Preparation of Bio-PSA/CO Patch

The Bio-PSA/CO patch was prepared by melt blending technique with the recipe as shown in Table 1. The STR 5L was first masticated with the two-roll mill (ML D6L12, Chareon Tut Co., Ltd., Samutprakarn, Thailand) under room temperature with different mastication time. The XG was then added into masticated STR 5L and then continuously mixed before adding CO at the end of the mixing process. The prepared NR-PSA was finally rolled and sheeted through the mill at room temperature. The obtained Bio-PSA/CO patches were coated onto backing materials (i.e., kraft paper, nylon film, PP film, and Al foil) with the same compressive pressure and kept under room temperature in a PP bag. The effect of mastication time (5, 10, and 15 min), nip gap (1, 2, 3, and 4 mm), and tack enhancer content (20, 40, 60, and 80 phr) on its adhesion properties were studied. The mixing times of XG and CO were fixed at 10 and 3 min, respectively.

### 2.4. Antimicrobial Activity Test

In this work, microbial spoilage of food can be detected by microbiological investigations. The banana cupcake was chosen as a bakery product to study because it is easily susceptible to mold spoilage in a few days. To prepare the samples, a piece of the cake was placed in sealed side fold PP bags (7.5 × 18.0 cm^2^) with and without the developed Bio-PSA/CO patch (2 × 2 cm^2^) (CO amount/patch = 8.0 μL). The food packs were stored at 30 °C with 75% room humidity (RH) for simulating the commercial distribution conditions. The cake sampling was subsequently done every day to determine the number of viable microbes on the cake surface using the drop-plate method [29]. In brief, the surface of the banana cupcake was swabbed with a sterilized cotton bud before placing into a distilled water tube of 10 mL. Tenfold serial dilutions of the sample stock were prepared by varying from 1 × 10^−1^ to 1 × 10^−6^ and the dispense of 20 µL were then dropped onto the agar media plates of NA (bacteria) and PDA (yeast and molds) before incubation under 37 °C for 1 day and 3 days, respectively.

### 2.5. Molecular Weight Determination

Molecular weight (Mw), number-average molecular weight (Mn), and polydispersity index (PDI) of the NR matrix in the obtained PSA patches were investigated by using gel permeation (GPC) gel perforation chromatography analyzer (GPC) (1260GPC/SEC MDS, Agilent Technologies, Santa Clara, CA, USA) with KF-806M and KF-803L columns. The PSA samples with a concentration of 0.001 g/mL were dissolved with tetrahydrofuran (THF) before filtrating by a 0.45 μm membrane needle. The THF was also applied as a mobile phase with a flow rate of 1.0 mL/min, at 40 °C. The refractive index detector was used as a signal detector.

### 2.6. Adhesive Properties Tests

#### 2.6.1. T-Peel Testing (Peel Strength of Bio-PSA/CO)

The Bio-PSA/CO (6.25 × 2.54 cm^2^) was fixed to several materials including kraft paper, nylon film, PP film, and Al foil with a size of 12.50 × 2.54 cm^2^. The samples were stored at room temperature for 24 h before conducting the test. The test was carried out by using a universal testing machine (Tinius Olsen H10KS, Tinius Olsen TMC-United States, Horsham, PA, USA) with a crosshead speed of 12.7 mm/min and 100 N load cell. Maximum forces of T-peel were reported according to the ASTM D3807 [20].

#### 2.6.2. Lap Shear Testing (Shear Strength of Bio-PSA)

The tests were set and conditioned similar to the peel strength determination. However, the Bio-PSA/CO and substrate sizes were used as 2.54 × 2.54 × 0.2 cm^3^ and 10.16 × 2.54 cm^2^, respectively. The testing distance was set at 2.54 cm relating to the length of the sticking area. Shear strength was then expressed as the shear force per unit area of the testing sample, and calculated according to the ASTM D5868–01 [20].

### 2.7. Morphological Analysis

Morphology of the Bio-PSA/CO was determined by a scanning electron microscopy (SEM) (JEOL JSM-5800LV microscope, Tokyo, Japan). The specimens were fractured by soaking in liquid nitrogen and the fractured surface was sputter-coated with platinum. The SEM observation was carried out with an accelerating voltage of 20 kV.

### 2.8. Statistical Analysis

ANOVA was used as a statistical approach and executed by using Statistical Package for the Social Science for Windows (SPSS). Three samples were averaged for each treatment, except molecular weight determination. The significant difference and mean values were obtained by Duncan’s test at a confidence level of *p* < 0.05.

## 3. Results and Discussion

### 3.1. Antimicrobial Activity

Figure 1 shows the physical appearance and antimicrobial activity of the banana cupcake samples in PP bag with and without the NR-PSA/CO patch for 9 days under 30 °C, 75% RH. The results demonstrated that the total microbial counts of the cupcake products decreased from 5,293 CFU/cm^2^ to 15 CFU/cm^2^ for bacterial and decreased from 10,501 CFU/cm^2^ to 11 CFU/cm^2^ for yeasts and molds at days 9. Moreover, the molds no longer appeared on the cupcake surface when observed by the naked eye. This was plausible due to the vapor of cinnamaldehyde which is the main component of CO can inhibit the growth of food spoilage microorganisms by destroy the cell wall structure and decrease the metabolic activity of the cell according to the report of Zhang et al. (2015) [30]. Other types of essential oil such as oregano oil [2,5,31], clove oil [4,6], garlic oil [10], and turmeric oil [32,33] could also use as an alternative natural antimicrobial substance depending on the type of microorganisms that cause the deterioration of the food. Moreover, the NR-PSA patch could be used as substrate materials for the controlled release of the CO into the headspace of food and extended the product shelf-life. However, the main adhesion properties between the NR-PSA/CO patch and a wide range of available packaging materials were the key factors that must be concerned and further examined for applying the patch into antimicrobial food packaging.

### 3.2. Effect of Mixing Conditions on Properties of NR-PSA/CO

#### 3.2.1. Influence of Nip Gaps

In order to develop the NR-based PSA, the reduction in molecular weight NR was the main factor that affected the adhesion properties of the PSA [18,19,20,21]. Therefore, the effect of nip gaps has been first investigated by varying in the range 1–4 mm. The NR-PSA/CO production was carried out with 20 phr of XG + 0.06 phr of CO in a mastication time of 10 min. The adhesion properties of the NR-PSA/CO prepared with different nip gap on kraft paper, nylon film, PP film, and Al foil were shown in Table 2. The trend of increased peel and shear strengths of the patches with decreasing nip gaps was observed. However, converting the NR-PSA/CO using nip gaps of 3 and 4 mm exhibited extremely poor adhesion properties and the test results could not be observed. This was due to the reduction in applied shear force during mixing by increasing the nip gap between the rolls, leading to the formation of rubber with large molecular weight. Therefore, it was difficult to adhere to the patches on the substrates by a mechanical-interlocking mechanism. In order to clarify and confirm the above explanation, Mn, Mw, and PDI of the NR prepared from different nip gaps were determined and the results are shown in Figure 2. The developed NR-PSA with different nip gaps (1 and 2 mm) was selected to test in order to obtain distinctly different results. It could be seen that the values of Mw and Mn of NR decreased from 1.7 × 10^5^ and 3.5 × 10^5^ g/mol to 1.6 × 10^5^ and 3.0 × 10^5^ g/mol, respectively when the nip gap was decreased from 2 to 1 mm. Moreover, the PDI values were also ostensibly decreased with decreasing the nip gap, indicating that the polymer chain was more monodisperse [16,34]. However, at a nip gap of 1 mm, the samples showed 100% cohesive joint failures after peeling. It means that there was residual adhesive on the substrates, while the cohesive failure no longer appeared when the nip gap was changed from 1 to 2 mm as shown in Figure 3. This phenomenon indicated that the dramatic reduction in the Mw of NR-PSA/CO with low NR chain entanglement reduced the cohesive strength as a result of weak interaction between the polymer chains.

Among the series of substrates, kraft paper exhibited the highest adhesion properties of the NR-PSA/CO (Table 2). This could be explained by the roughness of the substrate surface (Figure 4a) with more contact area between the adhesive and the paper. The low molecular weight of NR molecule could easily penetrate into the porous surface of the substrate and hence the higher bonding forces [35], while the non-porous materials (i.e., nylon, PP, and Al foil) (see Figure 4b–d) exhibited lower adhesion properties. The proposed model of adhesion mechanism of mechanical interlocking for NR-PSA/CO with smooth and non-smooth surfaces of substrates was presented in Figure 5. However, the other surface properties such as surface energy and glossiness of the substrate materials might also affect the adhesion properties of NR-PSA/CO.

#### 3.2.2. Influence of Mastication Time

To find out the optimal processing condition for the adhesive patches without cohesive failure, the effect of mastication time on the adhesion properties and failure was investigated. The adhesion properties of NR-PSA/CO produced from 20 phr of XG + 0.06 phr of CO are presented in Table 3. The nip gap of 2 mm for NR mastication was found to be a suitable gap that avoided the cohesive failure of PSA after peeling. An increase of mastication time led to an increase in the T-peel of NR-PSA/CO for kraft paper, nylon, and Al foil (Table 3). The results suggested that the considerable reduction in the molecular weight of NR during mastication enhanced the adhesion properties of PSA [16,18,19,20]. Figure 5 shows that there was no residue of NR-PSA/CO appears on all the coating substrates after peeling. Consequently, the results imply that the molecular weight of NR was not too low and the level of NR entanglement was high enough for resisting the breaking down of intermolecular bonding from the given adhesive substance. However, the decrease of shear strength was observed in the case of NR-PSA with kraft paper by increasing the mastication time from 5 to 10 min.

### 3.3. Effect of XG Content On Properties of NR-PSA/CO

The influence of XG content (20–80 phr) on the adhesion properties of NR -PSA patches were studied. The results showed that the adhesion properties of NR-PSA/CO could be enhanced by the addition of XG as a tack enhancer. The peel strength of the PSA increased with increasing the XG content for all substrates as shown in Table 4. In general, bulk polymer chain relaxation occurs while attached to the materials and the adhesive layer are present on the removal of it [35]. Therefore, this phenomenon results in decreasing the mobility of the polymer molecules up on the addition of XG inducing friction of the chain slippage [36] and providing high degree of molecular chain interlocking. As a result, the adhesion property might be also enhanced by inducing stronger intermolecular interactions [20] among the hydrophilic group of XG molecule and the substrate material like kraft paper as proposed in Figure 6. Nevertheless, the reduction in peel strength could be revealed when the XG content was over the maximum limit. The addition of greater XG causes an agglomerated rigid structure on the surface/near surface region of NR-PSA/CO. This results in hardening the surface and reduction of the tack energy of PSA on the material surface. Moreover, agglomeration of XG might also reduce the strength of a material leading to decrease the polymer-tack enhancer interaction and caused cohesive failure (Figure 6). There was no considerable difference in the shear strength of NR-PSA when the XG content increased from 20 phr to 80 phr. However, the appearance of cohesive joint failure could be seen in case of NR-PSA/COs at higher XG level.

## 4. Conclusions

The active natural rubber-based pressure-sensitive adhesive (NR-PSA/CO) for antimicrobial applications was successfully produced by a single-step process and the XG could be used as a tack enhancer. The developed NR-PSA/CO could be applied as a sticky patch for an active packaging system with different packaging materials. The CO released from the patches into the packaging headspace caused prolonging the shelf-life of the banana cupcake. Changes in the processing conditions (i.e., nip gap and mastication time) and XG content affected the adhesion properties and cohesive strength of the NR-PSA/CO. The nip gap distance of 2 mm with a mastication time of 15 min was a suitable condition for the NR-PSA/CO preparation. However, the maximum adhesion properties of the PSA could be obtained with XG content varied in a range of 40–60 phr, depending upon the coated substrate material.

## Figures and Tables

**Figure 1 polymers-13-00199-f001:**
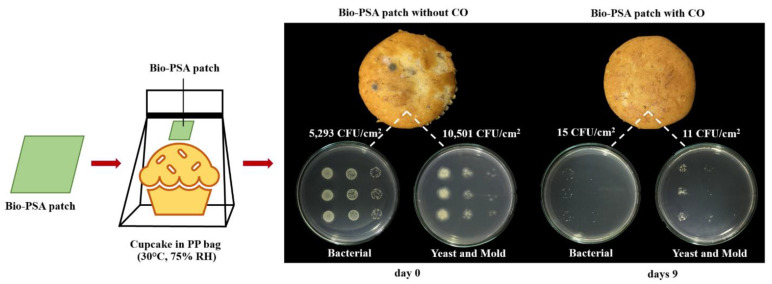
Physical appearance and antimicrobial activity of banana cupcake samples in a polypropylene (PP) bag with and without the natural rubber (NR)-PSA/CO patch at days 9 under 30 °C, 75% room humidity (RH).

**Figure 2 polymers-13-00199-f002:**
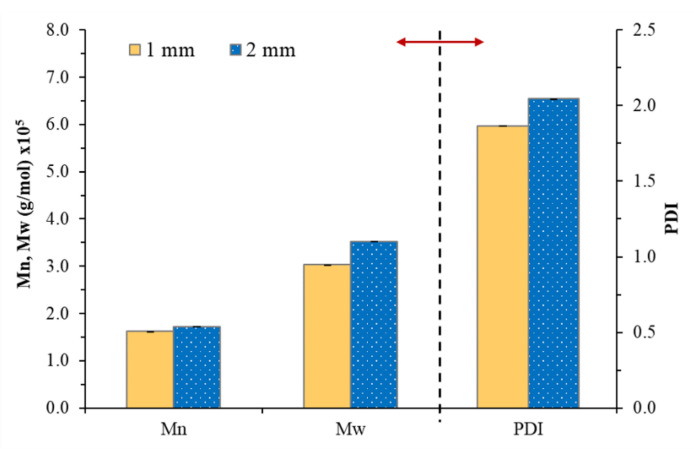
The Mw, Mn and polydispersity index (PDI) of NR obtained from NR-PSA/CO with 20 phr of XG + 0.06 phr of CO with a mastication time of 10 min for different nip gaps (1 and 2 mm).

**Figure 3 polymers-13-00199-f003:**
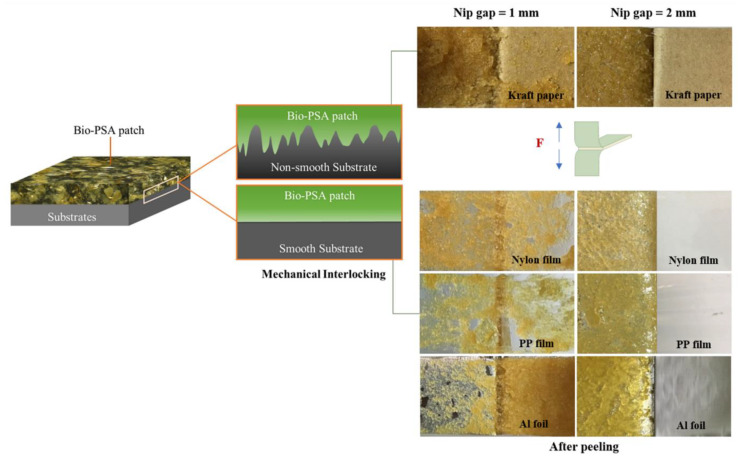
Adhesion model of mechanical interlocking in case of NR-PSA/CO with smooth and non-smooth surfaces along with the physical appearance of the substrate surfaces after peeling. NR-PSA/CO was produced using 20 phr of XG + 0.06 phr of CO with a mastication time of 10 min for 1 and 2 mm nip gaps.

**Figure 4 polymers-13-00199-f004:**
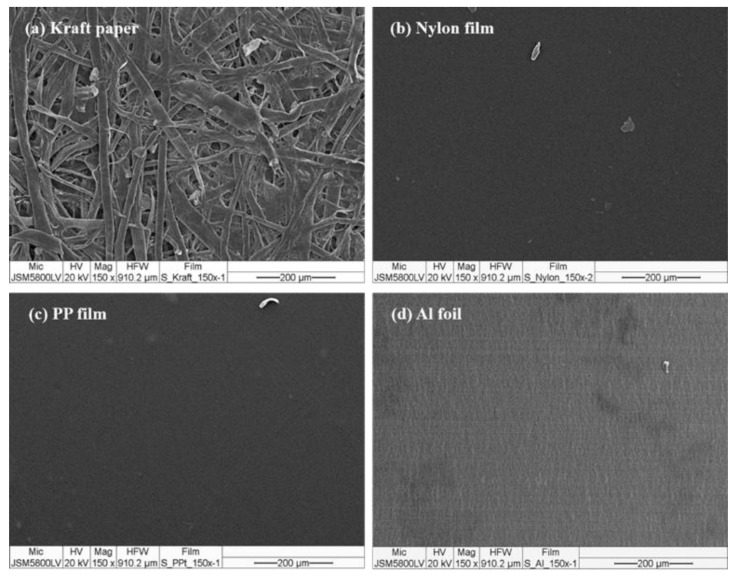
SEM micrographs of the substrate surfaces: (**a**) kraft paper; (**b**) nylon film; (**c**) PP film, and (**d**) Al foil.

**Figure 5 polymers-13-00199-f005:**
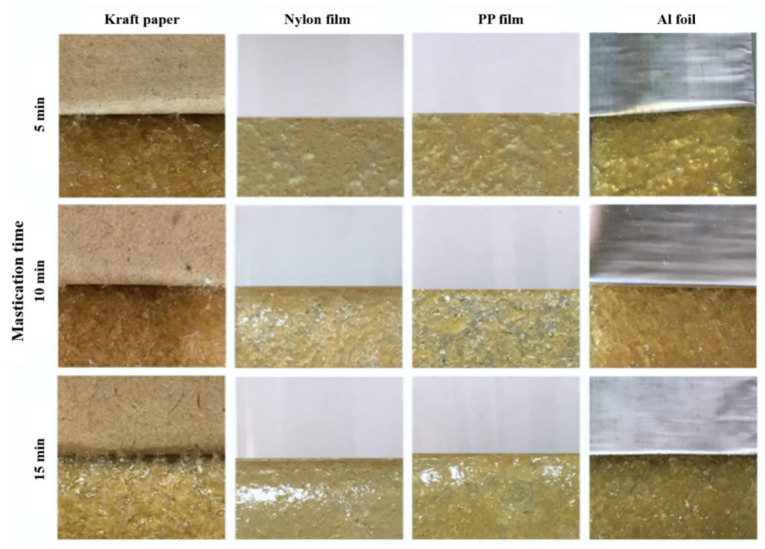
Physical appearance of NR-PSA/CO produced with 20 phr of XG + 0.06 phr of CO by using 2 mm nip gap with different mastication time (5–15 min).

**Figure 6 polymers-13-00199-f006:**
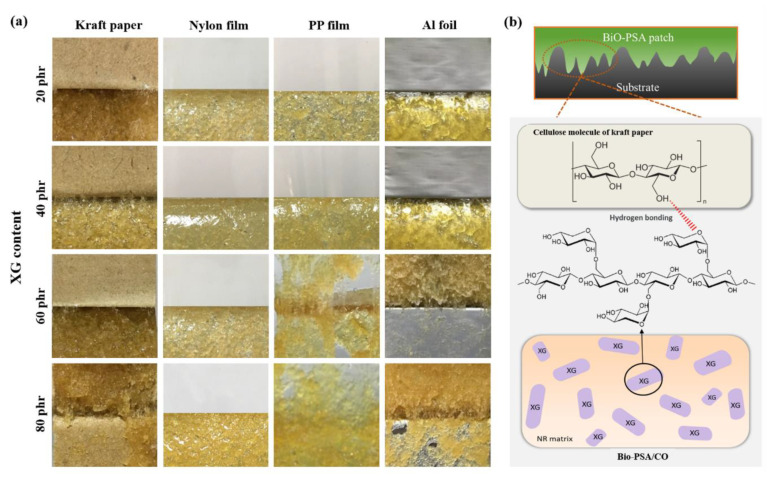
(**a**) Physical appearance of the NR-PSA/CO produced with 2 mm nip gap at a mastication time of 15 min with different XG content (20–80 phr); (**b**) proposed model of intermolecular interaction among cellulose molecules and XG of PSA.

**Table 1 polymers-13-00199-t001:** Recipe and mixing conditions for the bio-based pressure-sensitive adhesives patch incorporating cinnamon oil (Bio-PSA/CO) patches preparation.

Materials	Weight (phr)
STR 5L	100
Xyloglucan (XG)	20, 40, 60, and 80
Cinnamon oil (CO)	0.06 (1,600 μL) *

* The optimum amount that can inhibit the growth of microorganisms (gram-positive and gram-negative bacteria as well as yeasts and molds) was obtained from a previous investigation by a vapor diffusion method.

**Table 2 polymers-13-00199-t002:** Peel and shear strengths of NR-PSA/CO produced with 20 phr of XG + 0.06 phr of CO with a mastication time of 10 min for different nip gaps.

Nip Gap	Peel Strength (×10^2^ N/m)				Shear Strength (×10^4^ N/m^2^)			
(mm)	Kraft	Nylon	PP	Al Foil	Kraft	Nylon	PP	Al Foil
1	1.99 ± 0.11 ^bD^ *	0.83 ± 0.02 ^bA^ *	1.00 ± 0.08 ^bB^ *	1.14 ± 0.01 ^bC^ *	5.99 ± 0.28 ^bD^	4.30 ± 0.08 ^cB^	2.63 ± 0.10 ^cA^	5.28 ± 0.13 ^cC^
2	1.75 ± 0.10 ^aD^	0.03 ± 0.01 ^aA^	0.56 ± 0.01 ^aB^	0.94 ± 0.06 ^aC^	5.49 ± 0.46 ^bD^	3.50 ± 0.05 ^bB^	2.35 ± 0.09 ^bA^	4.11 ± 025 ^bC^
3	ND	ND	ND	ND	0.24 ± 0.03 ^aA^	0.33 ± 0.02 ^aB^	0.37 ± 0.05 ^aB^	0.36 ± 0.03 ^aB^
4	ND	ND	ND	ND	ND	ND	ND	ND

ND = non-detect. * Cohesive joint failures. Values are expressed as mean ± SD. Different small letter superscripts in the same column and capital letter superscripts in the same row indicate the significant difference (*p* < 0.05).

**Table 3 polymers-13-00199-t003:** Peel and shear strengths of NR-PSA/CO produced with 20 phr of XG + 0.06 phr of CO by using a two-roll mill mixer with a nip gap of 2 mm at various mastication time (5–15 min).

Mastication Time (min)	Peel Strength (×10^2^ N/m)				Shear Strength (×10^4^ N/m^2^)			
	Kraft	Nylon	PP	Al Foil	Kraft	Nylon	PP	Al Foil
5	1.51 ± 0.10 ^aC^	ND	0.49 ± 0.02 ^aA^	0.86 ± 0.02 ^aB^	9.08 ± 0.94 ^bC^	3.45 ± 0.17 ^aA^	2.48 ± 0.13 ^cA^	4.74 ± 0.95 ^abB^
10	1.75 ± 0.10 ^bD^	0.30 ± 0.01 ^aA^	0.56 ± 0.01 ^aB^	0.94 ± 0.01 ^aC^	5.49 ± 0.46 ^aD^	3.50 ± 0.05 ^aB^	2.35 ± 0.09 ^abA^	4.11 ± 0.25 ^aC^
15	2.14 ± 0.13 ^cC^	0.49 ± 0.03 ^bA^	0.59 ± 0.07 ^bA^	1.38 ± 0.13 ^eB^	5.15 ± 0.18 ^aC^	3.95 ± 0.46 ^bB^	2.28 ± 0.03 ^aA^	5.36 ± 0.35 ^cC^

ND = non-detect. Values are expressed as mean ± SD. Different small letter superscripts in the same column and capital letter superscripts in the same row indicate the significant difference (*p* < 0.05).

**Table 4 polymers-13-00199-t004:** Peel and shear strengths of NR-PSA/CO produced with 2 mm nip gap at a mastication time of 15 min with different XG content (20–80 phr).

XG Content (phr)	Peel Strength (×10^2^ N/m)				Shear Strength (×10^4^ N/m^2^)			
	Kraft	Nylon	PP	Al Foil	Kraft	Nylon	PP	Al Foil
20	2.14 ± 0.13 ^aC^	0.49 ± 0.03 ^aA^	0.59 ± 0.07 ^aA^	1.38 ± 0.13 ^aB^	5.15 ± 0.18 ^aC^	3.95 ± 0.14 ^aB^	2.12 ± 0.07 ^aA^ *	5.36 ± 0.35 ^aC^ *
40	5.64 ± 0.16 ^dB^	1.33 ± 0.17 ^cA^	1.77 ± 0.28 ^bA^	1.66 ± 0.35 ^aA^	8.24 ± 0.49 ^bD^	4.87 ± 0.21 ^bB^	2.97 ± 0.21 ^bA^	5.92 ± 0.52 ^aC^
60	4.55 ± 0.46 ^cC^	1.36 ± 0.14 ^cA^	2.44 ± 0.20 ^cB^ *	2.69 ± 0.20 ^cB^ *	8.51 ± 0.24 ^bD^ *	6.37 ± 0.49 ^cC^	2.89 ± 0.20 ^bA^ *	5.77 ± 0.08 ^aB^ *
80	3.76 ± 0.23 ^bC^ *	1.04 ± 0.15 ^bA^	2.17 ± 0.09 ^cB^ *	2.25 ± 0.01 ^bB^ *	9.50± 0.28 ^cD^ *	6.56 ± 0.24 ^cC^	2.97 ± 0.09 ^bA^ *	5.93 ± 0.52 ^aB^ *

* Cohesive joint failure. Values are expressed as mean ± SD. Different small letter superscripts in the same column and capital letter superscripts in the same row indicate the significant difference (*p* < 0.05).

## Data Availability

Not applicable.

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
