# Peer review of "Active Bio-Based Pressure-Sensitive Adhesive Based Natural Rubber for Food Antimicrobial Applications: Effect of Processing Parameters on Its Adhesion Properties"

_polymers, 2021, doi:10.3390/polym13020199_

Round 1

Reviewer 1 Report

The manuscript is well-written and can be considered for publication after addressing the following concerns:

1-In the introduction, the authors have talked about essential oils as antibacterial agents. Since plant extracts and nanoparticles have extensively been used as antibacterial agents, it is suggested to discuss them in the introduction section. Some suggested references are but not limited to: International journal of biological macromolecules, 2018, 109, 1219-1231; International journal of biological macromolecules, 2018, 115, 227-235; Colloid and Interface Science Communications, 2019, 33, 100211; Colloid and Interface Science Communications, 2020, 35, 100252; Food Packaging and Shelf Life, 2020, 23, 100435 

2- The characterization of the films should be improved by adding FT-IR, Oxygen penetration, water penetration/absorption, and XRD.

3- All the labels in the images are not clear. Please re-write all the labels in all figures/images. 

4- The quality of the images and schemes are too low. Please improve them. 

5- The discussion for antibacterial/antifungal data is week. Please elaborate more that how your film is acting against these strains. 

-Stay safe. 

Author Response

Reviewer I

Comments and Suggestions for Authors

The manuscript is well-written and can be considered for publication after addressing the following concerns:

1-In the introduction, the authors have talked about essential oils as antibacterial agents. Since plant extracts and nanoparticles have extensively been used as antibacterial agents, it is suggested to discuss them in the introduction section. Some suggested references are but not limited to: International journal of biological macromolecules, 2018, 109, 1219-1231; International journal of biological macromolecules, 2018, 115, 227-235; Colloid and Interface Science Communications, 2019, 33, 100211; Colloid and Interface Science Communications, 2020, 35, 100252; Food Packaging and Shelf Life, 2020, 23, 100435

Response 1:

  • Thank you for your suggestion. However, the suggested references are not related to this work. The antibacterial agents that can be applied for this application should be a volatile compound that can release its vapor into the headspace of close system packaging.

2- The characterization of the films should be improved by adding FT-IR, Oxygen penetration, water penetration/absorption, and XRD.

Response 2:

  • The main aim of this work is to study the effect of processing conditions on the tacking properties of the antimicrobial pressure-sensitive adhesive (PSA patch). Therefore, the film characterizations as mention above may not necessary for our work.

3- All the labels in the images are not clear. Please re-write all the labels in all figures/images.

Response 3:

  • The labels in the figures have been check and revised.

4- The quality of the images and schemes are too low. Please improve them.

Response 4:

  • The quality of the figures has been improved and all Figures have been replaced.

5- The discussion for antibacterial/antifungal data is week. Please elaborate more that how your film is acting against these strains.

Response 5:

  • More discussion has been added in the revised version of the manuscript (line #177–181, page #4).

Reviewer 2 Report

Why actually cinnamon oil was selected and why only one content of 0.06 phr was investigated? Any previous works or results? Could Authors propose other potential materials?

Regarding peel strength with respect to different materials Authors could refer to their chemical structure and potential interactions, not only to the roughness of surface. Maybe provide some schemes of potential interactions considering chemistry?

Very interesting work, but I think that except the mechanics, for the proposed application it is crucial that the material will release the desired portion of active compounds to the headspace. Therefore, Authors definitely should state why actually cinnamon oil was selected and why in proposed content. Moreover, the release should be investigated with respect to the processing conditions.

Author Response

Reviewer II

Comments and Suggestions for Authors

1. Why actually cinnamon oil was selected? why only one content of 0.06 phr was investigated? Any previous works or results? Could Authors propose other potential materials?

Response 1:

  • Cinnamon oil was selected due to its high antimicrobial efficiency as mentioned in the introduction of the manuscript. However, more information has been added in lines #78–80, page #2.
  • 0.06 phr of cinnamon oil was selected due to it is the optimum concentration that can inhibit the growth of microorganisms (gram-positive and gram-negative bacteria as well as yeast and molds) which was previously investigated by a vapor diffusion method. This information has been included in the revised manuscript (line # 120–122, page #3).
  • Other substances that can be used for this application have been proposed in the revised manuscript (line #180–182, page #4) and some references have been added.

2. Regarding peel strength with respect to different materials Authors could refer to their chemical structure and potential interactions, not only to the roughness of surface. Maybe provide some schemes of potential interactions considering chemistry?

Response 2:

  • Thank you for your suggestion. We are slightly sure that the peel strength of the PSA mainly comes from the roughness of the substrate. This can be explained by the obtained results which show that the polarity difference (chemical interaction) between kraft paper (hydrophilic) and the PSA (hydrophobic) is not the main factor affecting the adhesion properties. Although the -OH group of XG might interact with the -OH group of cellulose of kraft paper as described in the manuscript. For other substrate materials (i.e., Nylon, PP, and Al), they could not interact with the NR-based PSA when considering their chemical structures. Therefore, the addition of the interaction into the manuscript could make the reader confuse and lesser focus the main discussion point.

3. Very interesting work, but I think that except the mechanics, for the proposed application it is crucial that the material will release the desired portion of active compounds to the headspace. Therefore, Authors definitely should state why actually cinnamon oil was selected and why in proposed content. Moreover, the release should be investigated with respect to the processing conditions.

Response 3:

  • Cinnamon oil was selected as an active compound due to it is one of the essential oil that has excellent antimicrobial properties for many bacterial and molds as described in the manuscript. More details have been added in lines #78–80, page #2.
  • The release of cinnamon oil has been studied but the results will be reported in the further manuscript. And the results showed that the PSA patch can be used as a bio-device for a controlled release system of essential oil.